# Using Mixed Reality (MR) to Improve On-Site Design Experience in Community Planning

**Yuze Dan** [1,2]**, Zhenjiang Shen** [1,2,]*** , Yiyun Zhu** [3] **and Ling Huang** [4]

1   Division of Environmental Design, Kanazawa University, Kanazawa 920-1192, Japan; danyuze@outlook.com
2   Fujian Science & Technology Innovation Laboratory, Fuzhou 350108, China
3   School of Smart Health, Chongqing College of Electronic Engineering, Chongqing 401331, China; yyun401@163.com
4   School of Architecture and Urban planning, Chongqing University, Chongqing 400030, China; hwawa1025@vip.163.com
*   Correspondence: shenzhe@se.kanazawa-u.ac.jp; Tel.: +86-076-234-4650

**Abstract:** In recent years, designing in existing environments has been consistently emphasized in community planning. However, practicing such on-site design is not easy for designers, because the current technical conditions do not allow virtual design objects into real environments for 3D visualization and interaction. Thus, designers' intuitive design perceptions, accurate design judgments, and convenient design decisions are hardly supported. This paper explores the possibilities of using mixed reality (MR) technology to improve designers' on-site design experiences in community planning. For this, we introduced an MR design support system (MR-DSS) for the interactive on-site 3D visualization of virtual design objects. With the MR-DSS, we performed a design experiment with sixteen participants in a typical on-site design scene of community planning. The results showed that the MR technology could provide designers with intuitive design perceptions, accurate design judgments, and convenient design decisions, thus effectively improving their on-site design experiences.

**Keywords:** MR-DSS; on-site design intuitiveness; on-site design accuracy; on-site design convenience; community planning

## 1. Introduction

With the gradual fading of urban growth supremacism, community planning that focuses on optimizing urban stock assets has stepped onto the stage of history [1]. Compared to traditional urban planning and design, focusing on new areas of development, community planning has to confront the limitations originating from existing environments [1,2]. Further, current community environments generally contain some trivial details, such as old buildings, public facilities, street furniture, and landscapes, which are difficult to measure and integrate accurately into conventional basic design drawings, making it difficult for designers to make design judgments and decisions [3]. As a result, designers have gradually chosen to return to the design site to deeply perceive and understand the environmental details and to imagine and develop design proposals on-site [4]. Such an on-site design method originates from the "back to the things themselves" concept of the phenomenological movement [5], which encourages designers to immerse their proposals in community environments in order to evoke the spirit of the place [6].

However, limited by technicalities, the current on-site design of community planning mainly relies on the visual imagination of the designers [7], which is too abstract to support their intuitive design perceptions, accurate design judgments, and convenient design decisions. It therefore always leads to tedious, repetitive work [8].

In order to address the limitations relating to design intuitiveness, accuracy, and convenience, scholars have been committed to introducing the rapid development of

computer graphics (CG). Virtual reality (VR), a typical computer-generated imagery technology, was the earliest technology of this kind to be applied to the urban planning and design fields [9]. By assembling individual physical devices such as personal computers (PCs), head-mounted displays (HMDs), display screens, cameras, and sensors, early VR design support systems (VR-DSS), such as Geo walls, vision domes, and CAVEs, generated immersive virtual environments indoors, which allowed designers to make intuitive spatial judgments and design imaginations [10]. Meanwhile, 3D digital design objects were rendered for visualization in immersive virtual environments [11]. Improvements in computer performance provided the immersive virtual environments with more vivid details, including sound and animation [12]. The introduction of the tangible interactive interface allowed designers to interact with the virtual world in real time [13,14]. Nevertheless, the immersive virtual environments generated by VR were detached from the physical world, rarely providing real spatial perception [15]. Additionally, creating models for immersive virtual environments costs considerable time and effort and still might not restore all the real environmental details, offering scant support for an accurate design judgment [8]. In addition, the VR-DSS is generally involved in expensive and complicated devices that are usually operated in professional scientific laboratories and not available to most designers [14,16].

In summary, VR technology has the potential to improve on-site design intuitiveness, because it can simulate the vivid existing environments. However, the simulation of the existing environments would require a lot of effort and hardly contribute to improving design convenience. Additionally, the simulated virtual existing environments cannot support real spatial perception and thus can do nothing in terms of improving design accuracy.

Subsequently, the more advanced computer-generated imagery technology, augmented reality (AR), was introduced to the urban planning and design fields into compensate for the shortcomings of VR. AR can create environments where digital information can be inserted into a predominantly real-world view [17]. With portable devices, such as smartphones and mobile tablets, AR can bring virtual design objects into real design scenes for visualization [18,19]. Additionally, portable devices can be a tangible input interface to provide real-time manipulation behaviors (e.g., moving, rotating and zooming) for designers to adjust their virtual design objects in AR environments [4]. At the same time, real-time environmental analysis (such as wind, light and heat) for the virtual design objects, combined with remote computing, can be rendered in AR environments [8,20]. Since this 2D display on the screen of the portable device barely describes the real 3D spatial relationship, it was difficult for designers to make intuitive design perceptions, even if they returned to the design site [21]. Similarly, the input interface of the AR design support system (AR-DSS) on the screen of the portable device was relatively stiff. These interactions did not support flexible design actions, and it was difficult for designers to make accurate judgments [21].

In short, depending on portable devices, AR technology can bring virtual design objects into real design environments, thus avoiding the repeated trips between the design studios and design site and providing possibilities relating to the improvement of the on-site design convenience. Nevertheless, the visualization and interaction of virtual design objects are limited by the 2D screen display of portable devices, which hardly improve on-site design intuitiveness and accuracy.

Mixed reality (MR) technology combines the advantages of both VR and AR [22] since it can merge real and virtual worlds to produce new visual environments where physical and digital objects coexist and interact in real time [23]. The rapid development of hardware, especially the advent of the Microsoft HoloLens (Microsoft, Redmond, WA), a head-mounted display, makes MR available for some professional fields. Until now, MR has been used in the medical field for 3D visualization in surgery planning [24,25], the aerospace field for the simulation of the space exploration environments [26], the tourism field for interactive acoustic and visual navigation [27], the engineering field to promote communication and safety in on-site construction [28,29], and in other fields.

Overall, these successful applications demonstrate some important technical characteristics of MR, such as 3D on-site visualization, natural control mode (based on gaze and gesture), accurate spatial mapping, and real depth perception, which offer the potential to improve the current on-site design intuitiveness, accuracy, and convenience.

This study intends to apply mixed reality (MR) to community planning to improve on-site design with regard to intuitive perception, accurate judgment, and convenient decisions. Thus, an on-site design experiment, using an MR design support system (MR-DSS) for a typical community planning and design scene, was performed to examine the effectiveness of MR.

## 2. Materials and Methods

To realize the study objectives, we selected an MR-DSS based mainly on Microsoft HoloLens and carried out an on-site design experiment with sixteen participants to assess the effectiveness of MR in improving the on-site design experience in community planning.

### 2.1. MR-DSS

#### 2.1.1. Hardware

We chose Microsoft HoloLens (Microsoft Corp., Redmond, WA, USA) as the main hardware device of the MR-DSS in this study. Microsoft HoloLens is an MR head-mounted display (Figure 1) with a weight of 579 g and a battery life of two-three hours [30]. It contains an Intel 32-bit processor, a custom-built Microsoft Holographic Processing Unit (HPU 1.0), 2 GB RAM, and 64 GB flash memory [30]. Using optical waveguide technology, a 3D hologram can be viewed through the widescreen stereoscopic lens within a user's viewpoint of native surroundings [31]. A depth camera and four environmental understanding sensors are used to scan and sense the physical environment, which provides the potential for interaction between the virtual and real worlds [30]. An inertial measurement unit (IMU) receives input orders from users [30]. Finally, a speaker and four microphones have been integrated to support two-way communication [30].

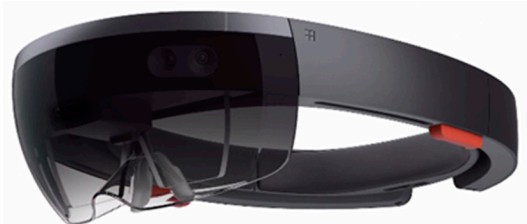

**Figure 1.** Microsoft HoloLens.

#### 2.1.2. Software

In this study, we used HoloDesigner, an MR application software on Microsoft HoloLens. HoloDesigner was developed in-house by the Unity 3D game engine in C# language, which could achieve the on-site interactive 3D visualization of design objects. As Figure 2 shows, HoloDesigner offers three interactive interfaces and six function buttons. Specifically, the three interactive interfaces are the model-loading interface, the material-loading interface, and the function selection interface. Six function buttons are involved in model adjustment, model placement, model removal, material selection, distance measurement, and interface switching. The design data, referring to 3D models and 2D material maps, are stored and loaded from the cloud server. It is worth noting that HoloDesigner not only supports human–computer interaction based on the gaze and gesture of users, but also supports physical interaction between the virtual and real worlds.

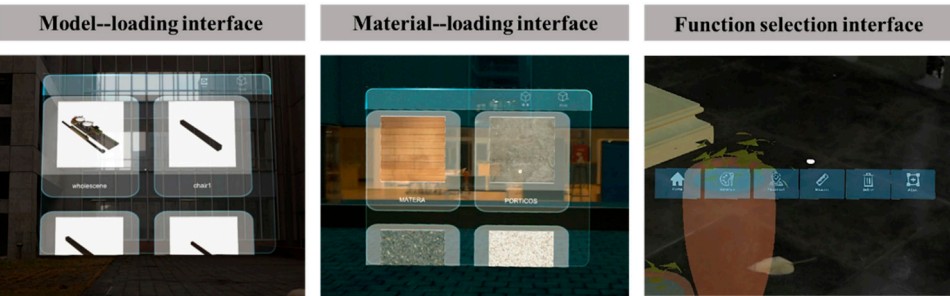

**Figure 2.** The interactive interfaces of HoloDesigner.

## 2.2. Case Selection and Data Preparation

We selected a typical public space design scene of community planning as the experimental case. This community space is located in Yuzhong District of Chongqing City in China, which was preliminarily designed by a professional group to add some benches, tree pools, and flowerpots (Figure 3). However, the preliminary design proposal has not been implemented, because it was relatively rough and needed to be further developed on-site.

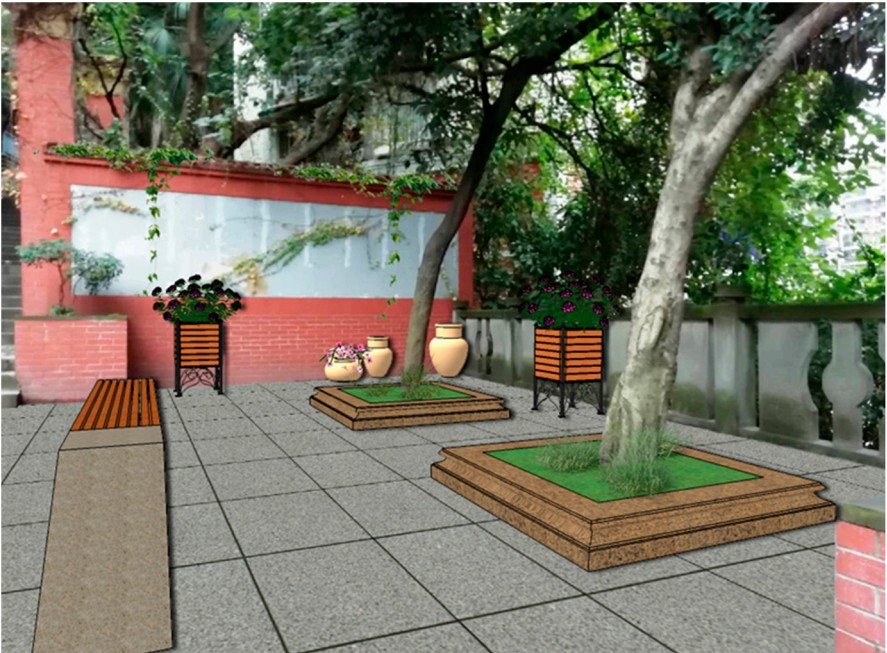

**Figure 3.** The preliminary design proposal for a typical public space of community planning.

According to the preliminary design proposal, the design data mainly contained 3D models and 2D material (wood and stone) maps of the planned benches, tree pools, and flowerpots. As shown in Figure 4, these 3D models and 2D material maps were created by Google Sketchup and Adobe Photoshop, respectively, and stored in the cloud server. Their uniform resource locators (URLs) were written in the designated data-implementing script of the MR-DSS to complete the data preparation.

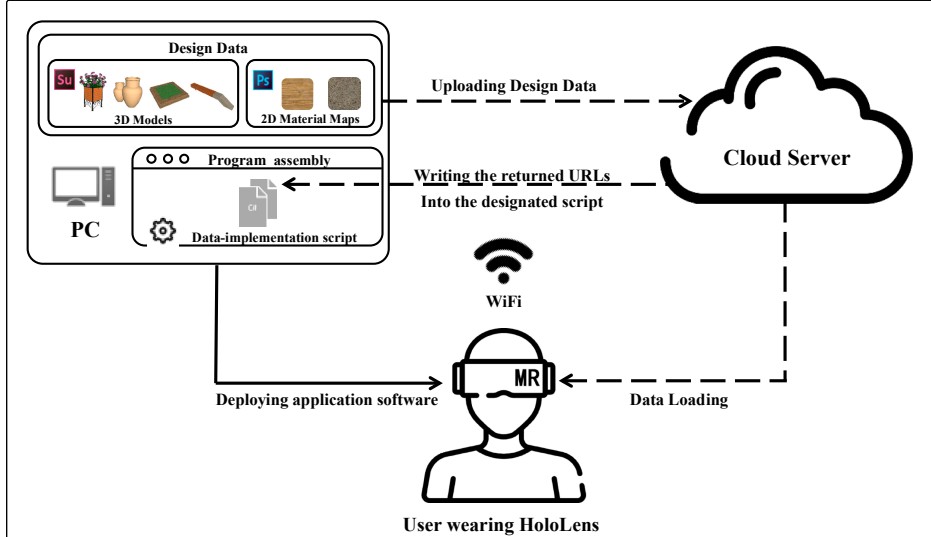

**Figure 4.** The design data preparation process.

### 2.3. On-Site Design Experiment

We recruited sixteen participants to take part in the on-site design experiment at the selected design scene. According to the study objectives, we evaluated the effectiveness of MR in improving the on-site design experience on the basis of three aspects (intuitiveness, accuracy, and convenience) through interviews and questionnaires.

#### 2.3.1. Participants

The sixteen participants included eight women and eight men, and the age range was 24 to 47 years (Mean $\pm$ SD: $30.38 \pm 7.18$ years, Median: 27 years). While only five had used MR before, all the participants were designers who had participated in relevant on-site design in community planning. Their design experience ranged from 3 to 26 years (Mean $\pm$ SD: $9.25 \pm 7.15$ years, Median: 6.5 years), which helped them to understand how MR changes their design experience.

#### 2.3.2. Experimental Procedures

First, as shown in Figure 5, we gave a ten-minute brief usage introduction of the MR-DSS to every participant separately (Figures 6–8).

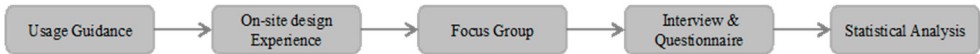

**Figure 5.** The on-site experimental procedures. Subsequently, each participant used the MR-DSS to engage in a ten-minute on-site design experience (Figure 6). According to the preliminary design proposal, each participant firstly selected the planned 3D street furniture models for on-site visualization in the real design scene and then made real-time design adjustments (suitable model, position, size, and materials), according to their real spatial perception and the community environmental details (Figures 7 and 8).

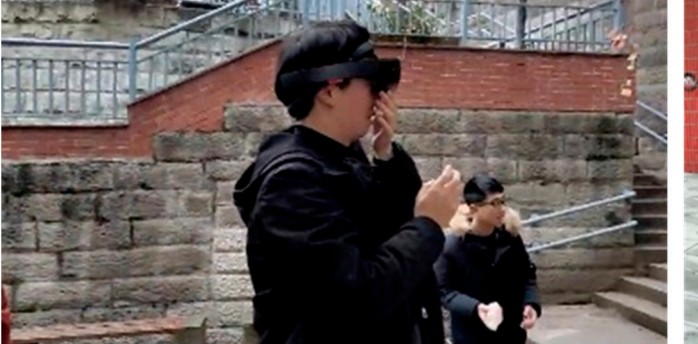

**Figure 6.** Participants engaged in the on-site design experience.

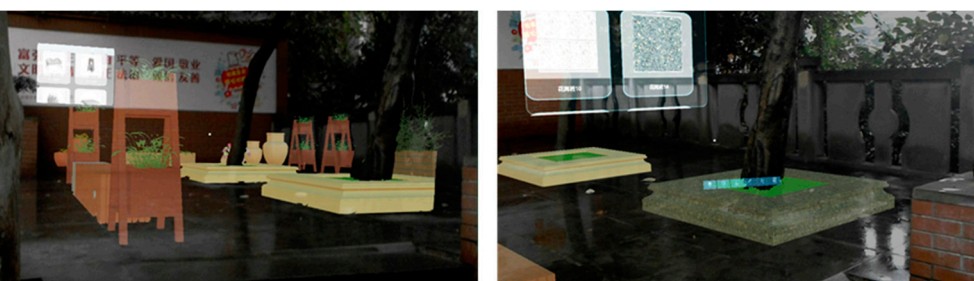

**Figure 7.** The 3D on-site visualization of the virtual design objects using MR.

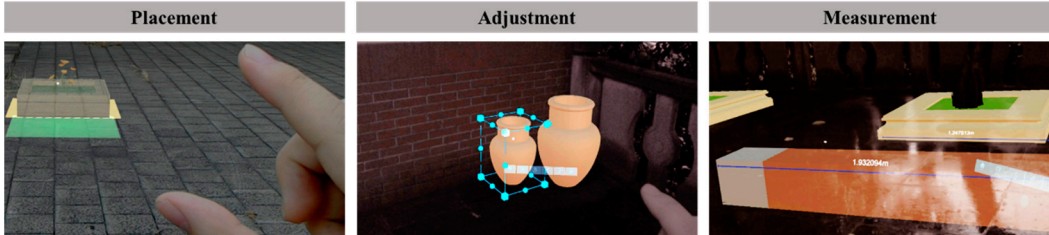

**Figure 8.** The real-time control for the virtual design objects using MR.

When the participants finished their on-site design experiences, we implemented a focus group to discuss the important factors for improving the on-site design experience using MR in terms of three aspects: intuitiveness, accuracy, and convenience.

Based on the focus group, we designed a questionnaire, and the responses to all the main questions were rated on a five-point Likert scale (Figure 9). Then, we conducted an interview with each participant using the questionnaire to examine the effectiveness of MR in improving the on-site design experience.

The results of the questionnaire are described in terms of the means and standard deviation. The Kruskal–Wallis test, with significance at the 5% level, was used to examine whether the questionnaire response trends differed based on the "design experience" and "MR usage experience" of the participants. Notably, for the Kruskal–Wallis test, we divided the participants into three groups (1–10 years, 11–20 years, and 21–30 years) according to their design experience and into two groups based on whether they had ever used MR devices. All the statistical analyses were performed using SPSS® Statistics Base 22 software.

**On-site design experience using MR**

**1 On-site design intuitiveness using MR**

**1.1 The realness of the physical world in MR environments:**

☐ ☐5 (Extremely real)      ☐ ☐4 (Very real)      ☐ ☐3 (Moderately real)      ☐ ☐2 (Slightly real)
☐ ☐1 (Not at all)

**1.2 The verisimilitude of the virtual design objects in MR environments:**

☐ ☐5 (Extremely verisimilar)      ☐ ☐4 (Very verisimilar)      ☐ ☐3 (Moderately verisimilar)
☐ ☐2 (Slightly verisimilar)      ☐ ☐1 (Not at all)

**1.3 The immersion of the MR environments:**

☐ ☐5 (Excellent)      ☐ ☐4 (Very good)      ☐ ☐3 (Good)      ☐ ☐2 (Fair)      ☐ ☐1 (Poor)

**2 On-site design accuracy using MR**

**2.1 The design materials selection using MR:**

☐ ☐5 (Extremely accurate)      ☐ ☐4 (Very accurate)      ☐ ☐3 (Moderately accurate)
☐ ☐2 (Slightly accurate)      ☐ ☐1 (Not at all)

**2.2 The design position judgment using MR:**

☐ ☐5 (Extremely accurate)      ☐ ☐4 (Very accurate)      ☐ ☐3 (Moderately accurate)
☐ ☐2 (Slightly accurate)      ☐ ☐1 (Not at all)

**2.3 The design size control using MR:**

☐ ☐5 (Extremely accurate)      ☐ ☐4 (Very accurate)      ☐ ☐3 (Moderately accurate)
☐ ☐2 (Slightly accurate)      ☐ ☐1 (Not at all)

**3 On-site design convenience using MR**

**3.1 The comfort of the MR-DSS:**

☐ ☐5 (Extremely comfortable)      ☐ ☐4 (Very comfortable)      ☐ ☐3 (Moderately comfortable)
☐ ☐2 (Slightly comfortable)      ☐ ☐1 (Not at all)

**3.2 The difficulty of learning the MR-DSS:**

☐ ☐5 (Extremely easy)      ☐ ☐4 (Very easy)      ☐ ☐3 (Moderately easy)      ☐ ☐2 (Slightly easy)
☐ ☐1 (Not at all)

**3.3 The control of the MR-DSS:**

☐ ☐5 (Excellent)      ☐ ☐4 (Very good)      ☐ ☐3 (Good)      ☐ ☐2 (Fair)      ☐ ☐1 (Poor)

**Figure 9.** The questionnaire with the main questions.

## 3. Results and Analysis

In this section, we describe and analyze the main results of this on-site design experiment based on our research goals: on-site design intuitiveness, on-site design accuracy, and on-site design convenience.

### 3.1. On-Site Design Intuitiveness Using MR

According to the interviews, we saw that the mean and standard deviation of the score for on-site design intuitiveness using MR were 4.15 and 0.21, respectively (as shown in Figure 10), which mainly included three concrete factor evaluations in our questionnaire.

For the realness of the physical environments, the mean and standard deviation of the score were 5.00 and 0.00 apart (Figure 10). Almost all the participants expressed that MR had not changed any details of the physical environment of this community public space; thus, they could experience completely real spatial perception even while wearing the MR head-mount display Microsoft HoloLens. Second, regarding the verisimilitude of the virtual design objects, the mean and standard deviation of the score were 3.81 and 0.40, respectively, as Figure 10 shows. Most of the participants stated that MR technology not only presented stereoscopic forms and exact scales for 3D street furniture models but also described life-like details for them. A few participants responded that there were some chromatic aberrations and sawtooth for 3D street furniture models, but this did not affect their perception and judgment. With respect to the immersion of the MR environments, the mean and standard deviation of the score were 3.63 and 0.50, respectively, as shown in Figure 10. Furthermore, most of the participants said that in the MR environments, they

could walk freely about the site to deeply understand the real environmental details and choose a suitable 3D street furniture model for the appropriate position, size, and material using gestures and gazes. Nevertheless, some participants pointed out that the relatively overloaded head-mount display and its narrow field of visualization (FOV) influenced their immersion experience to some extent in the experiment.

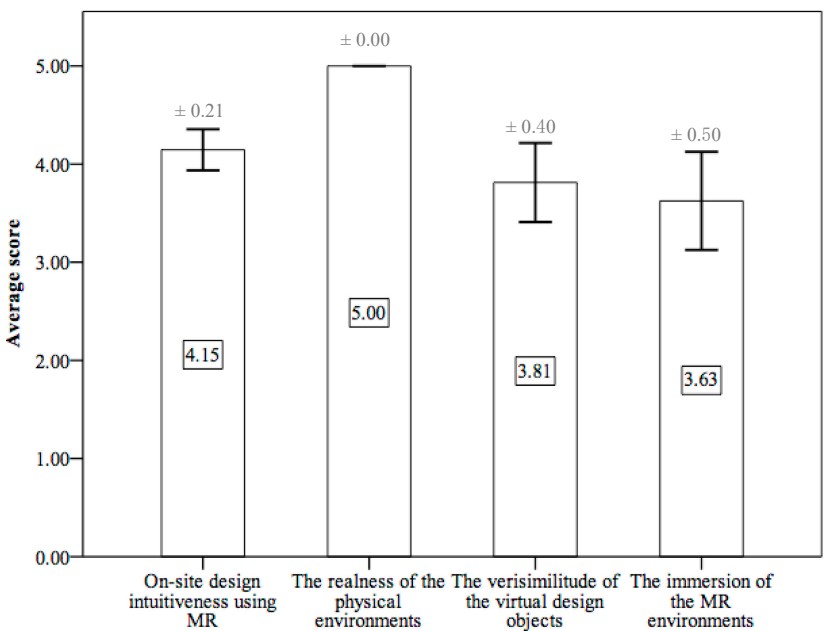

**Figure 10.** The mean and standard deviation of the score for on-site design intuitiveness using MR, the realness of the physical environments, the verisimilitude of the virtual design objects, and the immersion of the MR environments.

Through the Kruskal–Wallis test (Tables 1 and 2), we also found no significant differences in the scores of on-site design intuitiveness using MR in the different MR usage experience groups ($p = 0.112 > 0.05$) or in the different design experience groups ($p = 0.145 > 0.05$). We found that the improvements in on-site design intuitiveness originated from the natural technological characteristics of MR, regardless of the user.

**Table 1.** The Kruskal–Wallis test for different MR usage experience groups in on-site design intuitiveness using MR.

| | Mean Ranks of Scores for the MR Usage Experience Groups | | *p*-Value |
|---|---|---|---|
| | With MR usage experience | Without MR usage experience | (Kruskal–Wallis) |
| On-site design intuitiveness using MR | 11.00 | 7.36 | 0.112 |

**Table 2.** The Kruskal–Wallis test for different design experience groups in on-site design intuitiveness using MR.

| | Mean Ranks of Scores for the MR Design Experience Groups | | | *p*-Value |
|---|---|---|---|---|
| | 1–10 Years | 11–20 Years | 21–30 Years | (Kruskal–Wallis) |
| On-site design intuitiveness using MR | 9.17 | 9.50 | 3.00 | 0.145 |

### 3.2. On-Site Design Accuracy Using MR

From the interviews, we found that the mean and standard deviation of the score for on-site design accuracy using MR were 3.65 and 0.56, respectively, chiefly involving three concrete factor evaluations in the questionnaire (Figure 11).

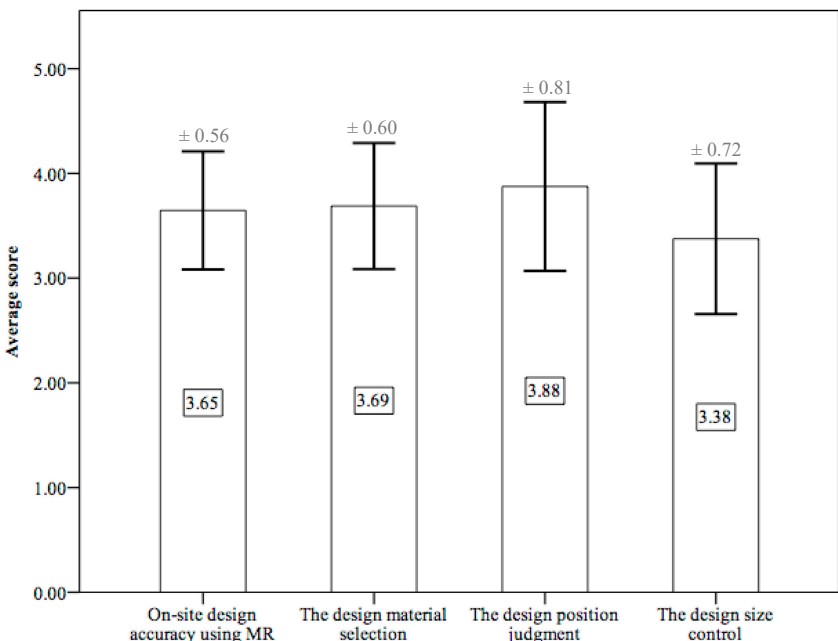

**Figure 11.** The mean and standard deviation of the score for on-site design accuracy using MR, design material selection, design position judgment, and design size control.

In terms of design material selection, the mean and standard deviation of the score were 3.69 and 0.60, respectively (Figure 11). Most participants stated that MR could render vivid textures and colors for various materials, which contributed to the relatively accurate material selection in specific existing community environments. A minority of participants, however, indicated that the rendering of the material was based on prepared 2D texture maps that could not represent the 3D details. Concerning design position judgment, the mean and standard deviation of the score were 3.88 and 0.81, respectively, as shown in Figure 11. The participants universally stated that MR displayed a totally real spatial relationship; thus, they could easily make position judgments about virtual design objects in the real community scene. With the measurement function of this MR-DSS, they were even able to acquire accurate spatial distances and coordinates. However, a small number of participants with rich design experience did not agree. They noted that they could still make accurate position judgments on site by deducing the planning and design layout in their minds, even without MR. Regarding design size control, the mean and standard deviation of the score were 3.38 and 0.72, respectively (Figure 11). Similar to the results for design position judgment, most of the participants reported that MR technology aided in effectively controlling the design size in this experiment, while those with rich design experience indicated that the effectiveness of MR was only mediocre. Furthermore, some participants responded that the unfamiliar interaction mode challenged them in accurately controlling the size of the virtual design objects in the experiment.

From the Kruskal–Wallis test (Table 3), we also saw significant differences ($p = 0.049 < 0.05$) in the scores for on-site design accuracy using MR between different MR usage experience groups. More precisely, the average rating (4.07) from the group with MR usage experience was significantly higher than that (3.45) from the group without MR usage experience, as shown in Figure 12. The Kruskal–Wallis test (Table 4) additionally revealed significant differences ($p = 0.045 < 0.05$) in the ratings of on-site design accuracy using MR among different design experience groups. Further, from Figure 13, we can see that the average scores for on-site design accuracy were 3.89, 3.53, and 2.83 in the 1–10 year, 11–20 years, and 21–30 years design experience groups, respectively, showing a clear downward trend with the increase in design experience.

**Table 3.** The Kruskal–Wallis test for different MR usage experience groups in on-site design accuracy using MR.

| | Mean Ranks of Scores for the MR Usage Experience Groups | | *p*-Value |
| --- | --- | --- | --- |
| | With MR usage experience | Without MR usage experience | (Kruskal–Wallis) |
| On-site design accuracy using MR | 11.90 | 6.95 | 0.049 |

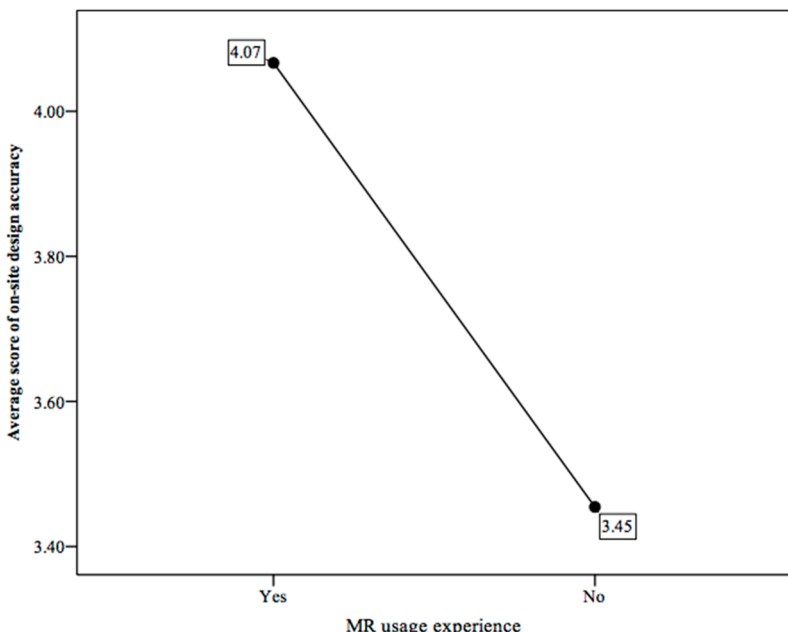

**Figure 12.** The average scores for on-site design accuracy using MR in different MR usage experience groups.

**Table 4.** The Kruskal–Wallis test for different design experience groups in on-site design accuracy using MR.

| | Mean Ranks of Scores for the Design Experience Groups | | | *p*-Value |
| --- | --- | --- | --- | --- |
| | 1–10 Years | 11–20 Years | 21–30 Years | (Kruskal–Wallis) |
| On-site design accuracy using MR | 10.56 | 7.50 | 1.75 | 0.036 |

*3.3. On-Site Design Convenience Using MR*

In accordance with the interviews, the mean and standard deviation of the score for on-site design convenience using MR were 3.90 and 0.36, respectively, which primarily contained three specific factor evaluations in our questionnaire (see Figure 14).

First, regarding the comfort of the MR-DSS, the mean and standard deviation of the score were 3.56 and 0.51, respectively (Figure 14). We found that some participants wore the head-mount display Microsoft HoloLens with their hands supporting the weight. However, most of the participants did not report feeling uncomfortable during their ten-minute design experience with MR. Only a small number of participants pointed out that the Microsoft HoloLens was uncomfortable for someone wearing glasses. Regarding the difficulty of learning the MR-DSS, the mean and standard deviation of the score were 4.44 and 0.51, respectively, as shown in Figure 14. Most of the participants stated they were able to grasp the basic manipulation rules of the MR-DSS, during the ten-minute usage introduction. The experiment also demonstrated that almost all the participants could engage in essential design actions with this MR-DSS, Regarding the control of the MR-DSS, the mean and standard deviation of the score were 3.69 and 0.48, respectively (Figure 14). We could see that almost all the participants had achieved real-time design control with

this MR-DSS. Nevertheless, most responded that they were not used to the current natural control mode based on gestures and gazes and used a large amount of effort in performing tiny detail manipulation.

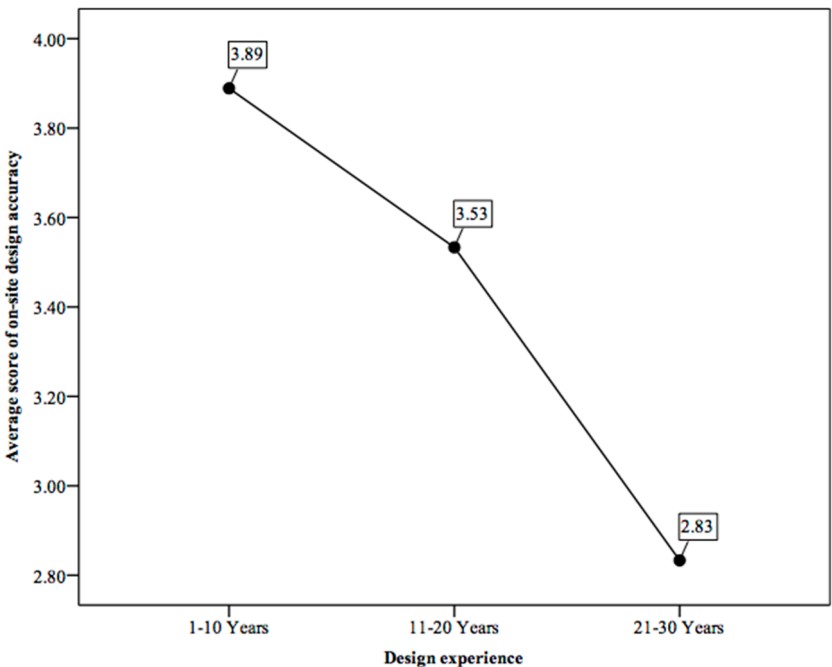

**Figure 13.** The average scores for on-site design accuracy using MR in different design experience groups.

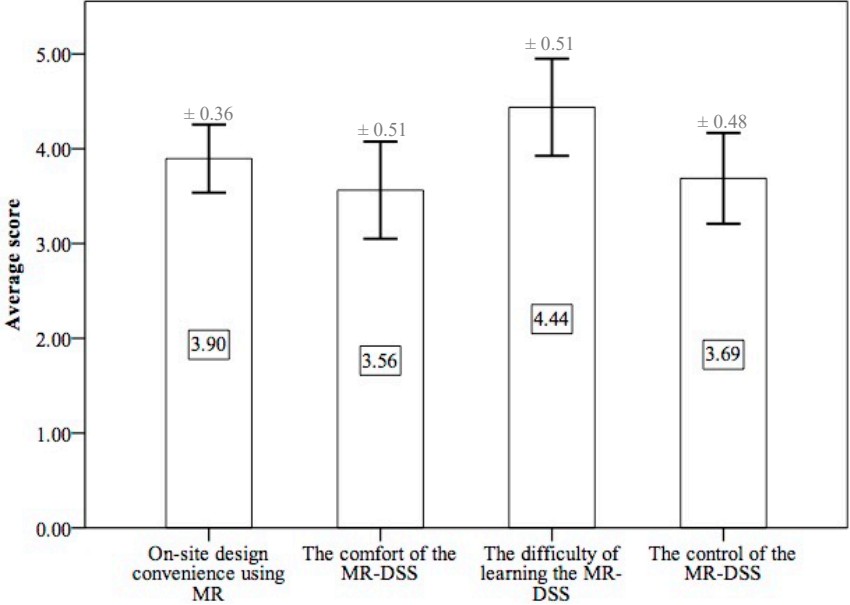

**Figure 14.** The mean and standard deviation of the score for the on-site design convenience using MR, the comfort of the MR-DSS, the difficulty of learning the MR-DSS, and control of the MR-DSS.

The Kruskal–Wallis test (Table 5) suggested significant differences ($p = 0.014 < 0.05$) in the scores of on-site design convenience with MR among different design experience groups. Specifically, the average scores in different design experience groups (1–10years, 11–20 years, and 21–30 years) were 4.11, 3.73, and 3.33, respectively, indicating a noticeable drop as design experience increased (Figure 15). Additionally, there were no identified

differences ($p = 0.077 > 0.05$) between the different MR usage experience groups with respect to the scores for on-site design convenience using MR, according to the Kruskal–Wallis test (Table 6). Most participants stated that the basic control of this MR support system was easy to grasp, even those who had never used an MR device.

**Table 5.** The Kruskal–Wallis test for different design experience groups in on-site design convenience using MR.

| | Mean Ranks of Scores for the Design Experience Groups | | | *p*-Value |
|---|---|---|---|---|
| | 1–10 Years | 11–20 Years | 21–30 Years | (Kruskal–Wallis) |
| On-site design convenience using MR | 11.22 | 6.20 | 2.00 | 0.014 |

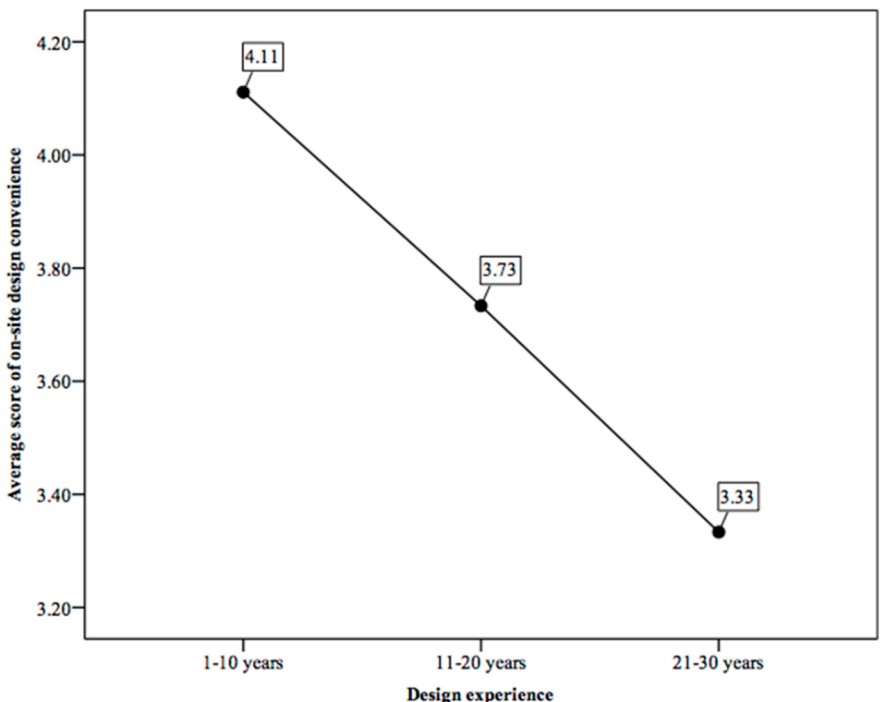

**Figure 15.** The average scores for on-site design convenience using MR in different design experience groups.

**Table 6.** The Kruskal–Wallis test for different MR usage experience groups in on-site design convenience using MR.

| | Mean Ranks of Scores for the MR Usage Experience Groups | | *p*-Value |
|---|---|---|---|
| | With MR usage experience | Without MR usage experience | (ANOVA) |
| On-site design convenience using MR | 11.50 | 7.14 | 0.077 |

## 4. Discussion and Conclusions

This study aimed to use mixed reality (MR) technology to improve on-site design experiences in community planning. To test its feasibility, we invited sixteen designers to participate in an on-site design experiment using MR for a typical community planning and design scene. The results validate that MR can significantly improve on-site design intuitiveness, accuracy, and convenience in community planning.

Specifically, MR can create interactive visualization environments where life-like 3D virtual design objects are displayed in the real world. In MR environments, designers can therefore not only understand the specific community environmental details but also visualize potential design objects and make further design deductions intuitively. MR

environments present a wholly real spatial relationship between virtual objects and the real world, which contributes to designers making accurate position perceptions and size judgments when imagining, deducting, and assessing their design proposals. At the same time, the vivid texture rendered by MR could aid designers in finding suitable design materials within the community contexts. With current portable MR-DSS and brief usage instructions, designers can acquire basic design control skills in MR environments. Thus, with MR, they can immerse their design proposals in an actual community scene and make real-time design adjustments conveniently instead of resorting to repeated observation, recording, sketching, and modeling between their studios and the design site.

Importantly, it can also be seen from the research results that although almost all the participants admitted that using MR led to improvements in on-site design of community planning, those with rich design experience gave relatively low ratings in the interviews. More specifically, the experienced designers had cultivated mature design habits that relied on their visual imagination through many years of planning and design practice. While they did experience the advantages of MR in this ten-minute on-site design experiment, it was difficult for the experienced designers to change long-standing design habits in such a short time. They preferred conventional on-site design methods that might be more tedious and complicated to the unfamiliar new technology. In their opinion, MR might serve as an auxiliary to current on-site design of community planning when necessary.

However, there are still some limitations to our study. First, regarding the hardware, we selected Microsoft HoloLens, which was the most advanced MR head-mount display at the time. Its technical characteristics (3D holographic visualization, natural control mode, accurate spatial mapping, and real depth perception) fit our study purposes well. Its weight (579 g), however, makes it unsuitable for wearing over long periods, and its low battery life (less than three hours) makes it less suitable for supporting sustainable outdoor design work. Additionally, the narrow FOV of the Microsoft HoloLens cannot present the whole MR environment, which can affect environmental perception and design judgment to some extent. Second, with regard to the software, because there were no suitable apps in the official Microsoft Store, we developed the application software, HoloDesigner, in-house. As we are not professional programmers, the current appearance, fluency, and stability of HoloDesigner need to be optimized further. This software also has only basic design functions (such as moving, zooming, and rotating), which barely support relatively complicated design work. Third, the design data used in the experiment were prepared beforehand, which might have limited the design imagination of the participants. Data preparation is a complex process that not only involves data creation, editing and uploading but also necessitates some coding work. If they are already familiar with the MR-DSS, designers can prepare the corresponding design data according to their specific design tasks by themselves. Fourth, in terms of the participants, some were from the same design institute. Thus, they might have already formed an attitude towards the new technology in their daily work before participating in the study, which might have affected their performance in the experiment. In addition, the sample size was relatively small; therefore, the results might not represent the attitudes of all designers. Finally, due to the simple experimental tasks and limited hardware power, the on-site design experience time with MR for a single participant was only ten minutes. Participants therefore might have found it difficult to fully understand the MR technology and make reasonable judgments about it.

Future work will examine how MR technology can support design communication between designers and residents in community planning. We intend to further improve the current MR-DSS to achieve a smoother operation, friendlier control, and richer functions, and, most importantly, multi-user sharing services. We then hope to apply it to facilitate the deliberation process of actual community planning projects.

**Author Contributions:** Conceptualization, Y.D. and Z.S.; methodology, Y.D.; software, Y.D.; validation, Y.D. and Y.Z.; investigation, Y.D. and Y.Z.; data curation, Y.D.; writing—original draft preparation, Y.D.; visualization, Y.D.; supervision, Z.S. and L.H. All authors have read and agreed to the published version of the manuscript.

**Funding:** This work was supported by the JSPS Grants-in-Aid for Scientific Research (No. 19K04750) and the National Natural Science Foundation of China (No. 51778078) and Fujian Science & Technology Innovation Laboratory for Optoelectronic Information of China (2021ZR139).

**Conflicts of Interest:** The authors declare no conflict of interest.

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
