# Peer review of "Using Mixed Reality (MR) to Improve On-Site Design Experience in Community Planning"

_applsci, doi:10.3390/app11073071_

Round 1

Reviewer 1 Report

This study investigates the improvement of designing experience for community public space using MR technology. The investigation may contribute to the understanding of the aspect related to the feasibility of new technology in design task. Although the topic of this manuscript is interesting, overall, the sections throughout the manuscript including the Abstract, Introduction, and, especially, the Method will need many significant improvements to be qualified for the standard of scientific rigorousness.

On the current state, I would suggest the authors to revisit the original study goal and its fitness to the method conducted. In addition, the authors may think again of what information to share from this study that can provide valuable information for the readers. In addition, I also recommend the authors to reference research articles published in the journal to assist the preparation of the manuscript.

I include some comments below that the authors may wish to consider:

  1. Abstract and Introduction: I suggest the authors to take another look at the content of the Abstract and the Introduction to let it tightly connect to the important information in this study. Currently, the information in Abstract and the arguments in Introduction need much work to improve.
  2. Major: p163-168: The tested factors” intuitive, accuracy, and convenience” are really the core of this study. However, those three factors are not explained and reviewed until their sudden show-up in the later part of the Method contained in a bracket. This is a serious flaw needed to be redeveloped and reorganized.
  3. Major: The study provides only 10 minutes for a participant in the experiment. And, there are 11 out 16 participant had no experience using MR. As a designing process includes many different phases, using this very limited time given to let the designers judge whether MR improves their professional designing task is somehow unreasonable and thus give an adversely impact on the study.
  4. Major: It seems that the experienced design experience in the study are to move the objects and to change sizes of the objects. However, those actions are really too limited to judge the effectiveness of MR in landscape design process.
  5. Major: Line 197-206: I strongly recommend the authors to develop several paragraphs for the instrument design. The instrument used has neither reference nor theory basis. Also, the instrument used does not really measure the “improvement” (the items do not have the ‘comparison’ aspect). This is an apparent issue of this study.      
  6. Major: L213-376: The analysis technique is another issue adversely impacting the study. In terms of a quantitative study, with the very small sample size of 16, it is not eligible for ANOVA analysis. The authors will need to look for other analysis techniques such as non-parametric technique, or increase the sample size. Please revisit the sampling consideration of ANOVA and the assumption tests of ANOVA for information.
  7. Line 31: The authors indicates that the designing environment is complicated, however, as that show in the study, the setting is relatively simple. This needs to be revised to meet with the characteristics of the study site.
  8. Because the terminology “MR design support system” are used a lot throughout the manuscript. I suggest the authors to use acronym or use dash to connect the words together to become one word to prevent confusion.
  9. L67: typo “n”
  10. I suggest the authors to use “design” rather than “planning” in this case as the study site is really small and the actions taken in the process does not really fit the “planning” concept.
  11. L98-102: This paragraph is not needed.
  12. All figures: Please move all the figures behind their texture explanations.
  13. Line 153-160: Please indicate what objects are made by Sketchup and what effects are made by Photoshop? You used material “maps” (line 157). What is the “map” you created here?

Reviewer 2 Report

The paper presents a very promising tool for the co-design process in Public space design. As an urban designer, I can not evaluate the "technological" aspects, but the paper presents clearly the research and application developed. I particularly found the final section highlighting the pro-cons of using these ICT innovative tools in co-design. 

Reading the paper, ideas came to mind. I write (below) some considerations (not necessary as additional parts to be included in the paper).
One aspect is that these "Mixed Reality" tools require (as underlined by Authors and architects interviewed) an advanced design definition to be applied. In my opinion, the co-design implies a community/stakeholders engagement since the first steps of the decision-making process. How can we "adapt" this ICT-MR tool (in my opinion, very promising for non-expert engagement and dialogue) to use these since the beginning steps of the design process (problem setting - strategic envisioning)? Is it possible to manage with the MR tool also the strategic dimension? OR the "interaction" by MR tools could be only referred/operate on "objects" (not on strategies or envisioning/imaginative ideas). 

The authors are proposing (as further application/development) to apply MR in planning. I think it could be more interesting and promising apply/test this MR tool into urban design and - primarily - to the urban regeneration processes due to the specific features of this tool in community and non -expert engagement (possibility to show possible scenarios as urban layouts, urban landscape).
I have some doubts about the possibility to manage the urban complexity with MR tools. The risk that the complexity could be "simplified" and the dialogues/co-design will focus/concern ONLY on the aspects of spatial or detailed design (only design alternatives and not in the definition of strategic scenarios - EG functions, synergies ...).

Reviewer 3 Report

Presented article with Title ”Using Mixed Reality (MR) to Improve the On-site Design Expe[1]3 rience in Community Planning” is writing on 16 pages with 15 figures, 6 tables, 0 equations and 31 references. This article is written clearly and comprehensibly. The article focuses on a practical example with a specific data. All data were verified oj sixteen participant. I think the questions were asked correctly but too general. The paper is worth publishing, but in my opinion, the manuscript content needs some improvements.

Suggestions:

  • Chapter 1. introduction is too short.
  • Chapter 2. is described too clearly and distinctly.
  • Please explain information value Figure 12, 13 and 15.

All the specific comments can be followed in reviewed copy of the manuscript.     

I recomend this paper publish in journal after revision.

Round 2

Reviewer 1 Report

Please use the full spelling to replace the use of abbreviations in keywords.

L107-111: Those statements seem that the authors have a presupposition and are about to do just an introduction. However, since this is a scientific and objective examination, please clearly and carefully setup your research questions.

L167-168: This is the comment in the last review (as follows) that remains no response and undeveloped in the literature review. The added lines (l42-45) only “mentioned” these three fore factors but not discussed nor defined.

“Major: p163-168: The tested factors” intuitive, accuracy, and convenience” are really the core of this study. However, those three factors are not explained and reviewed until their sudden show-up in the later part of the Method contained in a bracket. This is a serious flaw needed to be redeveloped and reorganized. “

Method: L197-199: Please develop the “survey design” sections. Currently, as the survey is the key measuring scale. I strongly recommend the authors to develop contents for the survey. The survey used has neither reference nor theory basis. Also, the instrument used does not really measure the “improvement” (the items do not have the ‘comparison’ aspect). This is an apparent issue of this study. This comment is in the last review but still remains undeveloped.

Results: Please note the number of participants in each analytical group (what is the n for each groups you compared?)

Because the study only has total 16 participants. As those participants are separate into different groups for comparison (like separate them into 3 age groups). Although you used non-parametric analysis, the sample size for each group is too small to make the analyses meaningful.   

In addition, the study did not mention about the comparisons among different age groups, and the experience groups. The gaps show in between the study design and the analyses.